# KNOWLEDGE TRANSFER FROM TEACHERS TO LEARNERS IN GROWING-BATCH REINFORCEMENT LEARNING

**Patrick Emedom-Nnamdi**[*]
Harvard University

**Abe Friesen**
DeepMind

**Bobak Shahriari**
DeepMind

**Nando de Freitas**
DeepMind

**Matt W. Hoffman**
DeepMind

## ABSTRACT

Standard approaches to sequential decision-making exploit an agent's ability to continually interact with its environment and improve its control policy. However, due to safety, ethical, and practicality constraints, this type of trial-and-error experimentation is often infeasible in many real-world domains such as healthcare and robotics. Instead, control policies in these domains are typically trained offline from previously logged data or in a *growing-batch* manner. In this setting a fixed policy is deployed to the environment and used to gather an entire batch of new data before being aggregated with past batches and used to update the policy. This improvement cycle can then be repeated multiple times. While a limited number of such cycles is feasible in real-world domains, the quantity and diversity of the resulting data are much lower than in the standard continually-interacting approach. However, data collection in these domains is often performed in conjunction with human experts, who are able to label or *annotate* the collected data. In this paper, we first explore the trade-offs present in this growing-batch setting, and then investigate how information provided by a teacher (i.e., demonstrations, expert actions, and gradient information—differentiated with respect to actions) can be leveraged at training time to mitigate the sample complexity and coverage requirements for actor-critic methods. We validate our contributions on tasks from the DeepMind Control Suite.

## 1 INTRODUCTION

Safe and reliable policy optimization is important for real-world deployments of reinforcement learning (RL). However, standard approaches to RL leverage consistent trial-and-error experimentation, where policies are continuously updated as new data is aggregated from environment interactions (Mnih et al., 2013; 2015; Silver et al., 2016; Van Hasselt et al., 2015). In this setting, agents intermittently act poorly (Ostrovski et al., 2021), often choosing to explore under-observed actions or act under substandard policies. In real-world settings, it is crucial due to ethical and safety reasons that an agent behaves above an acceptable standard during deployments. As such, recent work has focused on learning control policies either offline from previously gathered data or in a growing-batch manner, where at each deployment a fixed policy is used to gather experiential data and is updated using data aggregated from current and previous deployments (Lange et al., 2012; Agarwal et al., 2019; Levine et al., 2020; Gulcehre et al., 2021). Decoupling policy optimization from environment interaction in this fashion affords practitioners the ability to rigorously evaluate the performance and risks associated with the current policy between subsequent deployments (Gottesman et al., 2019; Thomas & Brunskill, 2016).

While learning in an offline or growing-batch manner allows for extensive pre-deployment evaluation, it is also known to hinder the agent performance (Ostrovski et al., 2021; Levine et al., 2020).

---

[*]Work done while at DeepMind. Correspondence to `patrickemedom@g.harvard.edu` or `mwhoffman@deepmind.com`

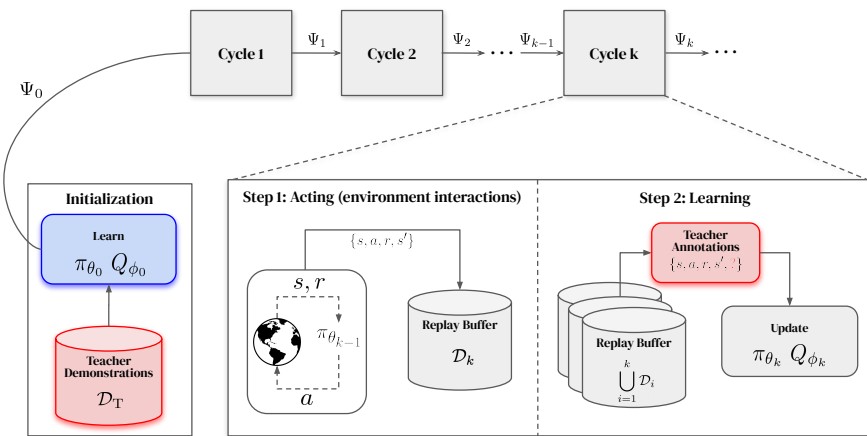

Figure 1: Growing-batch RL with teacher annotations. The policy and (optionally) critic networks are first initialized from offline data (Sections 3.2–3.3); in our work we use an offline dataset of teacher demonstrations. Within each cycle, a fixed policy $\pi_{\theta_{k-1}}$ is deployed within the environment and used to gather data in the replay buffer $\mathcal{D}_k$. Data from previous cycles are then aggregated (i.e., $\cup_{i=1}^{k}\mathcal{D}_i$) along with per-transition teacher annotations, and used to update the policy and critic networks, $\pi_{\theta_k}$ and $Q_{\phi_k}$, respectively. The forms of annotations are discussed in Section 3.4.

By deploying to the environment less frequently, these agents are often unable to eliminate the overestimation bias that exists with respect to out-of-distribution samples. As such, in comparison to agents trained via consistent online learning, offline and growing-batch agents are often deluded, exploring actions with overestimated value estimates (Fujimoto et al., 2019; Kumar et al., 2019; Siegel et al., 2020). Several remedies have been proposed to prevent this behavior including by preventing the value function from evaluating unseen actions (Peng et al., 2019; Wang et al., 2020; Kumar et al., 2020) or constraining the learned policy to remain close to the offline data (Nair et al., 2020; Fujimoto et al., 2019; Kumar et al., 2019). However, the performance of such agents is often limited by the quality and availability of batch data. While opportunities for additional data collection within the growing-batch setting can help alleviate these issues, they still tend to perform worse than their fully online counterparts due to limited coverage of the state-action space (Haarnoja et al., 2018; Liu & Brunskill, 2018). To mitigate these coverage requirements, pre-training has been extensively studied as a mechanism for obtaining good initial policies.

In this work, rather than training agents entirely from scratch, we first initialize policies in a supervised fashion via behavior cloning on teacher-provided demonstrations. Doing so naively, however, can result in a drop in performance when transitioning from pre-training to the growing batch setting, often due to poorly-initialized value functions (Uchendu et al., 2022; Kostrikov et al., 2021b;a). To avoid this we also investigate policy regularization to promote monotonic policy improvement across growing-batch cycles. This regularization aims to keep the current learned policy close to its pre-trained counterpart or to estimated policies from previous deployments during early periods of the agent's learning process. Additionally, in real-world domains, deployments are typically done in conjunction with human experts, who are able to annotate the collected data with additional information that can be leveraged at training-time to further improve policy optimization. These annotations can come in the form of demonstrations, alternative actions, or (weak) gradient information that are differentiated with respect to actions (and not parameters which are impractical to provide from an external system). In this work, we train agents in a growing-batch setting and investigate the benefits of incorporating realistic external information provided by *teachers* (e.g., another RL agent, a human, or a well-defined program) during policy improvement (see Figure 1). We investigate the use of transition-specific annotations from teachers, specifically considering (1) teacher-provided actions, where, for select transitions chosen via a value-based filter, we constrain the learned policy to remain close to the teacher's suggested action, and provide direct corrective information in the form of (2) teacher-provided critic gradients (differentiated with respect to actions), where the learned policy is nudged toward learning actions that maximize the teacher's internal (presumably unknown) representation of the value function.

We evaluate our proposed approach on a set of continuous control tasks selected from the DeepMind Control Suite using actor-critic agents trained under distributional deterministic policy gradients. Our results suggest that effective policy regularization paired with teacher-provided annotations offers a mechanism of improve the sample efficiency of growing-batch agents, while ensuring safe and reliable policy improvement between cycles.

## 2    BACKGROUND & GROWING-BATCH REINFORCEMENT LEARNING

Interactions between an agent and the environment can be modeled as an infinite-horizon Markov decision process (MDP) $(\mathcal{S}, \mathcal{A}, \mathbb{P}, R, \gamma)$ where at each time step $t$ the agent observes the state of the environment $s_t \in \mathcal{S}$, executes an action $a_t \in \mathcal{A}$ according to a deterministic policy $\pi$, and receives a reward $r_t = R(s_t, a_t)$. The goal of RL is to learn an optimal policy $\pi^*$ that maximizes the expected future discounted reward, or return, $G_{\pi^*} = \max_\pi G_\pi = \max_\pi E\left[\sum_{t=0}^\infty \gamma^t r_t | \pi\right]$ that it receives from the environment, where $\gamma \in [0, 1]$ is a discount factor.

We focus here on off-policy RL algorithms as these can learn from policies different than the current behavior policy, which is necessary when learning from older deployment cycles and expert data. We consider *growing-batch* settings where data generation is decoupled from policy improvement. That is, scenarios where updates to the policy are made only after large batches of experiential data are collected from the environment. We will also focus on actor-critic algorithms where $\Psi_k = \{\phi_k, \theta_k\}$ represent parameters used to model the agent's critic and policy networks, respectively. Under the growing batch setup, agent interaction with the environment and parameter updates are performed within structures we call cycles. Within a given cycle $k$, an agent interacts with the environment using the fixed policy $\pi_{\theta_{k-1}}$ and stores each transition within a replay buffer we denote as $\mathcal{D}_k$. Transitions gathered from previous cycles $\cup_{i=1}^{k-1} \mathcal{D}_i$ are then aggregated with $\mathcal{D}_k$ and are used to generate an updated $\psi_k$, i.e., an update of agent's model parameters learned via an off-policy learning algorithm. This process is generally repeated for a fixed number of cycles $N$ or until an optimal decision-making policy is retrieved.

While several parallels between online and growing batch RL can be drawn, our growing-batch experimental setup differs in two key aspects. Specifically, (1) the number of pre-specified cycles we explore is small, while (2) the size of each newly gathered batch dataset is large. This distinction is important when considering domains or areas of application such as clinical trials or deployments of self-driving vehicles, where performing nearly real-time continual parameter updates after collecting only a few transitions is impractical or, perhaps, infeasible due to safety, resource, and/or implementation constraints.

Under the actor-critic approach that we focus on in this work, we estimate a parametric policy $\pi_\theta$ by maximizing the expected return $\mathcal{J}(\theta) = \mathbb{E}_{(s,a)\sim\mathcal{D}}\left[Q^{\pi_\theta}(s, a)\right]$, where $Q^{\pi_\theta}$ is the associated value function. For continuous control tasks, $\mathcal{J}(\theta)$ can be directly optimized by performing parameter updates on $\theta$ with respect to the deterministic policy gradient:

$$\nabla \mathcal{J}(\theta) = \mathbb{E}_{s\sim\mathcal{D}}\left[\nabla_\theta \pi_\theta \nabla_a Q_\phi(s, a)\big|_{a=\pi_\theta(s)}\right]; \tag{1}$$

see (Silver et al., 2014) for further details on computing this gradient in practice. As is commonly done, we update the critic $Q_\phi$ by minimizing the squared Bellman error, represented under the following loss:

$$\mathcal{L}(\phi) = \mathbb{E}_{(s,a)\sim\mathcal{D}}\left[\left(Q_\phi(s, a) - (\mathcal{T}_{\pi_{\theta'}} Q_{\phi'})(s, a)\right)^2\right], \tag{2}$$

where we use separate target policy and value networks (i.e., represented under $\theta'$ and $\phi'$) to stabilize learning. For our set of experiments, we also make use of $n$-step returns in the TD-error from equation 2 and, rather than directly learning the value function, we use a distributional value function $Z_\phi(s, a)$ whose expected value $Q_\phi(s, a) = \mathbb{E}\left[Z_\phi(s, a)\right]$ forms our value estimate. For further details see (Barth-Maron et al., 2018), however alternative policy optimization methods could be used within our growing batch framework.

## 3  TEACHER-GUIDED GROWING-BATCH RL

### 3.1  ESTIMATING SAFE, RELIABLE, AND SAMPLE-EFFICIENT POLICIES

Real-world applications of RL typically have significant safety and ethical requirements that highlight the need for both (1) a good initialization of $\pi_0$ and (2) safe (and ideally) monotonic improvement of successive policies $\pi_k$ for each growing-batch cycle $k \geq 1$. Unfortunately, this is difficult to achieve when naively using value-based RL methods. For instance, while $\pi_0$ can be initialized from a batch of expert demonstrations using imitation learning techniques such as behavioral cloning (BC), a poorly-initialized state-action value function $Q^{\pi_0}$ can lead to a significant drop in performance when training a subsequent policy $\pi_1$, regardless of the initial quality of $\pi_0$ (Uchendu et al., 2022). While proper policy initialization can help reduce this drop, agents continually learning in a trial-and-error manner nevertheless risk encountering substandard intermittent policies (i.e., $G_{\pi_k} \leq G_{\pi_{k-1}}$).

To address these challenges, we make use of techniques that mitigate the risk of policy deterioration between cycles and hasten the learning process of conventional RL agents by leveraging external information at initialization and training time. Our approach takes advantage of queryable embodiments of knowledge we refer to as *teachers*. Teachers can provide well-informed knowledge of the task at hand in the form of the following:

1. demonstrations of the given RL process, or
2. training-time annotations (i.e., teacher-provided actions and gradient information—differentiated with respect to actions) of agent-generated transitions.

In what follows, we explore how agents can leverage these forms of knowledge for sample-efficient learning of policies within a stable and monotone training process.

### 3.2  POLICY INITIALIZATION

Rather than training agents using a randomly-initialized policy, we leverage demonstrations gathered from teachers, as is common in real-world RL. We assume transitions gathered by a teacher are of the format $\{(s, a, r, s')\}$ and are stored within the dataset $\mathcal{D}_{\mathrm{T}}$. Using batches of these transitions, we train an initial policy $\pi_0$ via behavioral cloning (BC)

$$\pi_0 = \arg\min_\theta \left( \frac{1}{|\mathcal{D}_{\mathrm{T}}|} \sum_{(s,a) \sim \mathcal{D}_{\mathrm{T}}} \|\pi_\theta(s) - a\|_2^2 \right).$$

We then initialize the value function $Q^{\pi_0}$ by performing policy evaluation with respect to $\pi_0$ (i.e., minimize $\mathcal{L}(\phi)$ in equation 2 using $\pi_0$).

Policies initialized using BC obtain baseline performance are comparable to the teacher for observations that closely resemble those gathered within the batch dataset $\mathcal{D}_T$. However, for out-of-distributions observations, BC-initialized policies tend to perform poorly due to compounding errors from the selection of sub-optimal actions (Ross et al., 2010). Furthermore, due to over-estimation bias for out-of-distribution observation-action pairs, optimizing BC-initialized $\pi_0$ by taking a few policy gradient steps according to equation 1 can essentially erase the performance gain from BC. This phenomena predicates the need for effective policy regularization procedures.

### 3.3  POLICY REGULARIZATION

To avoid forgetting the performance gain of the initialized policy during each subsequent policy optimization step, we explore augmenting the deterministic policy gradient loss with a *regularizer*, $\mathbb{E}_{s \sim \rho^\pi} \|\pi_0(s) - \pi_{\theta_1}(s)\|_2^2$. We generalize this for all cycles and obtain the following policy loss:

$$\mathcal{J}(\theta_k) = \mathcal{J}_{\mathrm{D4PG}}(\theta_k) + \lambda \underbrace{\mathbb{E}_{s \sim \rho^\pi} \|\pi_0(s) - \pi_{\theta_k}(s)\|_2^2}_{\text{BC regularizer}}, \tag{3}$$

where $\lambda$ is a regularization parameter. As each subsequent policy $\pi_{\theta_k}$ is trained, the BC-regularizer ensures that the learned policy remains close to the BC-initialized policy $\pi_0$ according to the

strength of regularization parameter $\lambda$. Due to the deterministic, continuous output of our policy we base this regularizer on the Euclidean distance between policy outputs, however for policies with stochastic outputs it would also be possible to make use of the Kullback-Leibler (KL) divergence $D_{\text{KL}}(\pi_0 \,\|\, \pi_{\theta_k})$ between $\pi_0$ and $\pi_{\theta_k}$.

However, while the BC-initialized policy serves as a good starting point, staying too close to it prevents the agent from improving on the expert behavior. We thus decay the strength of the regularizer over subsequent deployments by incorporating an exponential decay weight $\alpha \in (0,1)$ to the objective function in equation 3:

$$\mathcal{J}(\theta_k) = (1-\alpha)\, \mathcal{J}_{\text{D4PG}}(\theta_k) + \alpha\, \mathbb{E}_{s\sim\rho^\pi} \|\pi_0(s) - \pi_{\theta_k}(s)\|_2^2 ., \tag{4}$$

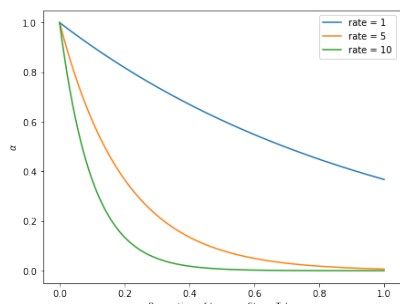

By treating $\alpha$ as a function of the total number of stochastic gradient (SGD) steps taken, we introduce a learning process that initially constrains the learned policy to remain close to the BC-initialized policy and gradually transitions to solely learning from the D4PG loss component. By choosing an appropriate rate parameter for $\alpha$, this form of regularization allows the learned policy to supersede the performance of $\pi_0$ as more data is gathered.

Figure 2: Regularization parameter $\alpha_n \in (0,1)$ as a function of $n$ learner steps taken, evaluated for various exponential-decay rates.

We also explore an alternative between-cycle regularizer that ensures that the learned policy $\pi_{\theta_k}$ stays close to the previously policy $\pi_{\theta_{k-1}}$, which allows the policy to adapt but slows the rate at which it does so:

$$\mathcal{J}(\theta_k) = (1-\alpha)\, \mathcal{J}_{\text{D4PG}}(\theta_k) + \alpha\, \mathbb{E}_{s\sim\rho^\pi} \|\pi_{\theta_{k-1}}(s) - \pi_{\theta_k}(s)\|_2^2. \tag{5}$$

As previously stated, our regularizer takes the form of the Euclidean distance between successive policies could be represented using the KL divergence for continuous, stochastic policies as done in Maximum a posteriori Policy Optimization (MPO) (Abdolmaleki et al., 2018).

### 3.4 TEACHER-PROVIDED ANNOTATIONS

While BC can provide good a starting policy on observed transitions, agents still run the risk of learning sub-optimal policies due to insufficient state-action coverage. We investigate the use of teachers to provide transition-specific annotations to improve sample complexity and facilitate optimization of the current policy $\pi_{\theta_k}$. The forms of possible annotations depend on the representation and accessibility of the teacher (i.e., our ability to query the teacher for advice). In our experiments, we represent our teacher as an RL agent with a deterministic policy that obtains either sub-optimal or optimal performance on a given task. The forms of annotations we consider include teacher-provided actions (i.e., $a^* \sim \pi^*(s)$) and critic gradients (i.e., $\nabla_a Q^*(s,a)$). Generally, teacher-provided actions function as expert demonstrations, while teacher-provided gradients serve as direct corrections for decisions made by the current policy. The practicality and accessibility of each of these forms of annotations are heavily dependent on the overarching use case and intended application at hand.

**Teacher-Action Annotations.** We first consider an annotation mechanism similar to DAgger (Ross et al., 2010). In imitation learning, DAgger introduces a mechanism for querying expert-suggested actions. These actions are used both at acting and training time to reduce the risk of compounding errors due to an induced distributional shift. Unlike DAgger, we only query the teacher's action during training time to provide transition-specific annotations for optimizing a reinforcement learning-based objective.

Specifically, we explore augmenting our agent's policy loss $\mathcal{J}_{\text{D4PG}}$ to include an $\ell_2$-loss component that evaluates the difference between the student's policy $\pi_{\theta_k}$ and the teacher-suggested action $a^*$ for each transition in $\mathcal{D}$. This constrains the agent's policy to remain close to the suggestions provided by the teacher. Furthermore, to encourage monotonic policy improvement between successive cycles of data aggregation and policy optimization, we utilize a between-cycle policy regularizer:

$$\mathcal{J}(\theta_k) = (1-\alpha)\, \mathcal{J}_{\text{D4PG}}(\theta_k) + \mathbb{E}_{s\sim\rho^\pi} \Big[ \beta_k \underbrace{\|\pi_{\theta_{k-1}}(s) - \pi_{\theta_k}(s)\|_2^2}_{\text{Between-cycle policy regularizer}} + \alpha \underbrace{\|a^* - \pi_{\theta_k}(s)\|_2^2}_{\text{Teacher-action component}} \Big]. \tag{6}$$

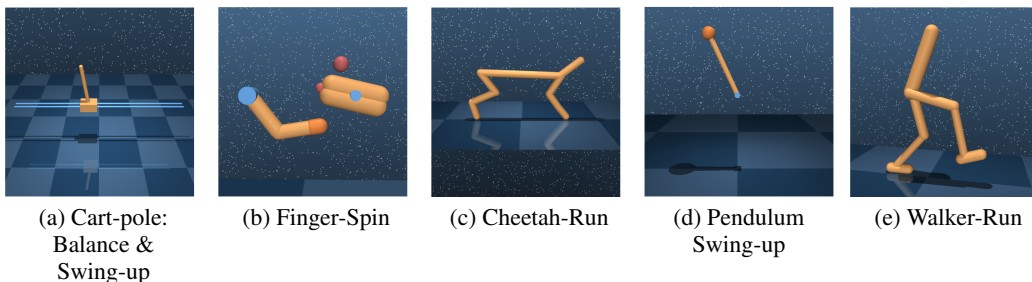

| (a) Cart-pole: Balance & Swing-up | (b) Finger-Spin | (c) Cheetah-Run | (d) Pendulum Swing-up | (e) Walker-Run |

Figure 3: The DeepMind Control Suite environments used in our experiments.

Under this objective, we also incorporate the exponential weight parameter $\alpha \in (0, 1)$ previously introduced in equation 4. Thus, as $\alpha$ decreases, our agent transitions from learning a policy that stays close to the teacher-suggested action to one that solely relies on the agent's conventional policy loss. Choosing an appropriate rate parameter $\alpha$ enables the learn policy $\pi_{\theta_k}$ to down-weight its reliance on the teacher's suggestions as more data is gathered.

In scenarios where evaluation is costly, choosing an appropriate parameter for $\alpha$ may be difficult. As such, we explore an adaptive procedure for selecting which policy loss component (i.e., D4PG vs. DAgger-like) to minimize at a per-transition basis. To do so, we introduce a Q-filter $\delta(s)$ and re-construct the deterministic policy gradient as

$$\nabla \mathcal{J}(\theta_k) = \mathbb{E}_{s \sim \rho^\pi} \left[ \underbrace{\left[1 - \delta(s)\right] \nabla_{\theta_k} \pi_{\theta_k} \nabla_a Q_{\phi_k}(s, a)\big|_{a = \pi_{\theta_k}(s)}}_{\text{D4PG component}} + \underbrace{\delta(s) \nabla_{\theta_k} \|a^* - \pi_{\theta_k}(s)\|_2^2}_{\text{Teacher-action component}} \right], \quad (7)$$

where $\delta(s) = \mathbb{1}[Q_{\phi_k}(s, a^*) \geq Q_{\phi_k}(s, \pi_{\theta_k}(s))]$ is an indicator function. Under this filtered approach, the agent's policy is optimized using the DAgger-like component only for transitions where the teacher-suggested actions obtain a value $Q_{\phi_k}$ that is larger than the value of the current policy $\pi_{\theta_k}$. This approach bears a resemblance to the optimization procedure introduced in the critic regularized regression (CRR) algorithm where advantage-weights are used to filter out actions that significantly deviate from the training distribution.

**Teacher-Gradient Annotations.** Additionally, for continuous control tasks, we consider directly incorporating teacher-provided gradient information $G_a(s, a)$ differentiated with respect to actions $a$. These gradients can be interpreted as the directions in which the agent should adjust their actions to enhance their current policy. This approach differs from using parameter-based gradients as corrective feedback since such information is challenging for an external system or teacher to supply. We envision several examples for using human feedback to estimate action-specific gradients. Some include:

1. A reward model that is learned from expert human feedback and differentiated with respect to actions, and

2. Aggregated human experiences that each provide (1) a preferential ordering suggesting directions each action should be moved towards and (2) a magnitude, indicating how much to move in the preferred direction.

In both examples, the gradient information prioritizes actions that aim to maximize the teacher's internal notion of a reward or value function (e.g., $G_a(s, a) = \nabla_a Q^*(s, a)$, where $Q^*$ is the teacher's value function). In general, we hypothesize that relying on teacher-provided gradients during the early periods of an agent's learning process can circumvent risks associated with learning from a poorly initialized or potentially over-estimated value function.

As a preliminary first step toward leveraging teacher-provided gradients, we augment the deterministic policy gradient introduced in equation 1 to include a component $G_a(s, a) \nabla_{\theta_k} \pi_{\theta_k}(s, a)$ that weights the current policy gradient according to gradient information provided by the teacher. As

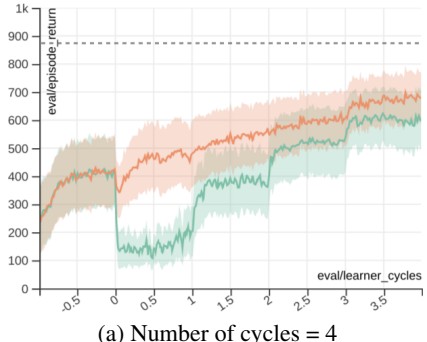
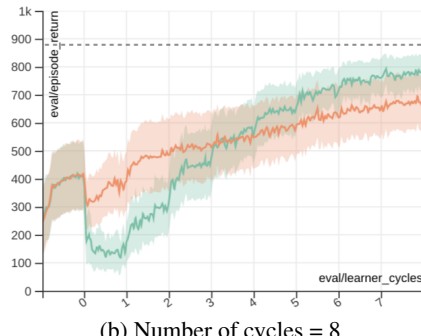

(a) Number of cycles = 4    (b) Number of cycles = 8

Figure 4: Episode returns averaged over all tasks (using 5 randoms seeds per task) for a varying number of cycles (under a fixed number of actor steps) comparing BC initialization only vs. BC-policy regularizer. The shaded regions represents the standard deviation across all tasks and random seeds. The dashed line represents the expert teacher's average baseline performance across all tasks.

such, we represent the augmented deterministic policy gradient as

$$\nabla \mathcal{J}(\theta_k) = \mathbb{E}_{s \sim \mathcal{D}} \Big[ (1 - \alpha) \, \nabla_{\theta_k} \pi_{\theta_k} \nabla_a Q_{\phi_k}(s, a) \big|_{a = \pi_{\theta_k}(s)} + \alpha \, \underbrace{G_a(s, \pi_{\theta_k}(s)) \nabla_{\theta_k} \pi_{\theta_k}(s)}_{\text{Teacher-provided gradient signal}} \Big] \qquad (8)$$

and incorporate the exponential decay weight $\alpha$ that decrements as functions of number of gradient steps taken. Thus, the agent leverages gradient information from the teacher primarily during early stages of its learning process.

## 4    EXPERIMENTS

We evaluate our series of methods on a set of continuous control tasks with observable states in the growing-batch setting. A number of these tasks also involve multidimensional actions spaces. Our results suggests that effective policy regularization paired with teacher-provided annotations works well in these challenging domains and serves to improve the sample efficiency of deterministic policy gradient algorithms, while encouraging monotonic policy improvement between cycles. Across all experiments, the total number of actor steps and learner steps are identical and fixed at specific values. Additionally, the number of cycles within each experiment can vary, with more cycles adding to diversity of data collected. Further details on our experimental setup are provided in Appendix A.2.

### 4.1    ENVIRONMENTS

DeepMind Control Suite (DSC) is a set of continuous control tasks used as a benchmark to assess the performance of continuous control algorithms. We consider the following six MuJoCo environments from the DSC: *cartpole-balance*, *cartpole-swingup*, *finger-spin*, *cheetah-run*, *walker-run*, and *pendulum-swingup*. An illustration of the environments is given in Figure 4. For our set of experiments, the dimensionality of the action space is low (i.e., $\leq 6$ degress of freedom). Additionally, a feature-based observation space is considered (i.e., no pixel-based observations under partial observability).

### 4.2    INVESTIGATED APPROACHES

We evaluate agents on the set of control tasks mentioned in section 4.1 and highlight the performance benefit of our proposed approaches (i.e., policy regularization and training-time annotations) against naively pre-training with behavioral cloning. Demonstrations from teachers performing each task were aggregated into datasets intended for pre-training. Each dataset contains 1M transitions from 1K episodes generated entirely from teachers. Additionally, annotations are queried by these same set of teachers at training time. Teachers were snapshots of RL agents that obtained either expert-level or mid-tier performance for each task. Expert-level snapshots achieved a per-episode return

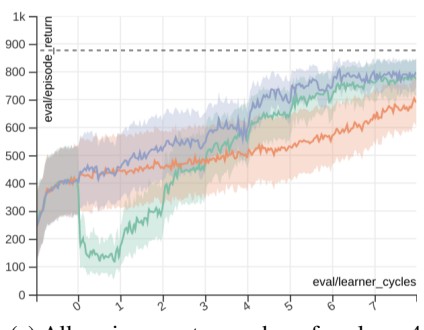

(a) All environments, number of cycles = 4

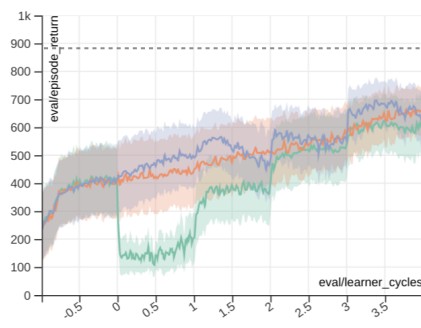

(b) All environments, number of cycles = 8

Figure 5: Episode return averaged over all tasks (using 5 randoms seeds per task) for varying number of cycles (under a fixed number of actor steps) comparing baseline with BC initialization only vs. BC-policy regularizer with decay rate = 1 and decay rate = 5. The shaded regions represents the standard deviation across all tasks and random seeds. The dashed line represents the expert teacher's average baseline performance across all tasks.

of roughly 900 (averaged over all tasks), and mid-tier snapshots achieved a per-episode return of roughly 600. To account for stochasticity in the training process, we repeat each training run 5 times with different random seeds.

During training, the agent's policy is evaluated every 100 learner steps in an identical environment under a different initialization. For each environment, these results are then averaged across each task and are displayed within each figure representing our main results. Note that pre-training with expert data does not guarantee an expert-level policy because key transitions that helped the expert agent achieve a high-reward state may be under-observed within the demonstration dataset (e.g., in cart-pole swing-up, an expert agent is able to immediately swing the pole up and then keep the pole upright for the vast majority of the episode).

**Policy Regularization.** In the the growing-batch setting, we compare BC pretraining with and without policy regularization. As shown in Figure 4, BC pre-training helps the agent achieve decent baseline performance across the environments on average. However, as the agent proceeds to learn according the D4PG objective functions, we observe a stark initial decline in performance as anticipated in Section 3.2. Policy regularization helps mitigate this issue by constraining the learned policy to remain close to the BC initialized policy. Here a regularization strength of 0.5 was used for the BC regularizer. While policy regularization performs well, we notice that (1) the drop in performance after pre-training is not completely eliminated and that (2) too much regularization prevents the agent from surpassing baseline performance as the number of cycles (and thus the number of times the agent is able to interact with the environment) increases.

To address these challenges, we examine the benefit of using exponentially decreasing regularization weights. Under this approach, we set $\alpha = 1$ in equation 4 and gradually decrease it following an exponential decay as the number of learner steps increases. The parameter $\alpha$ eventually reaches 0 once the allotted number of learner steps has been taken. In Figure 5, we notice that the dip in performance after pre-training is no longer present. As such, we observe that constraining the agent's policy to remain close to the BC initialized policy for the early stages of its learning process can help ensure a monotonic performance increase after pre-training. Additionally, as the number of cycles increases, the agent's policy is able to surpass the baseline performance for faster decay rates. While this approach works well, it requires hyper-parameter tuning to find a suitable value for the exponential decay rate.

**Teacher-Action Annotations.** We incorporate teacher-action annotations provided at training-time (paired with policy regularization) to examine how additional external information serves to further improve policy improvement between cycles. For teacher-action policy loss in equation 6, the between-cycle regularization parameter $\beta_k$ was set to 5, while the decay rate for $\alpha$ was separately chosen to be 1 and 5. Figure 6a highlights that, throughout the learning process, using teacher-action annotations out-performs solely relying on policy regularization after pre-training. This insight is observed for both choices for $\alpha$. However, the teacher-action policy loss exhibits no-

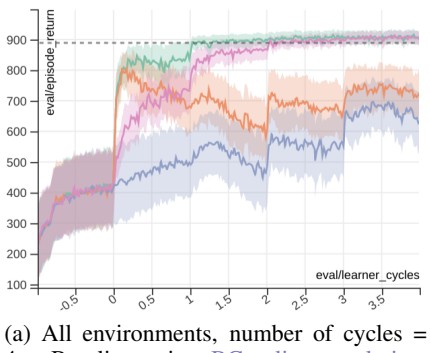

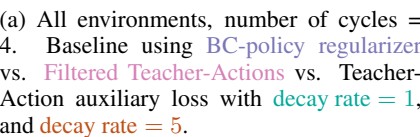

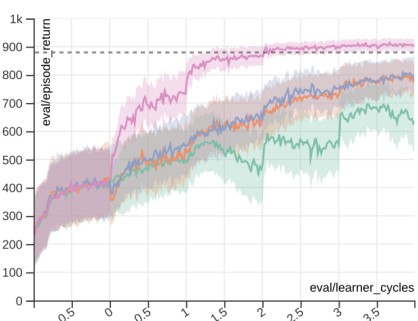

(a) All environments, number of cycles = 4. Baseline using BC-policy regularizer vs. Filtered Teacher-Actions vs. Teacher-Action auxiliary loss with decay rate = 1, and decay rate = 5.

(b) All environments, number of cycles = 4. Baseline using BC-policy regularizer vs. Filtered Teacher-Actions vs. Teacher-Gradient auxiliary loss with decay rate = 1, and decay rate = 5.

Figure 6: Episode return averaged over all tasks (using 5 randoms seeds per task) under teacher-provided annotations. The shaded regions represents the standard deviation across all tasks and random seeds. The dashed line represents the expert teacher's average baseline performance across all tasks.

ticeable sensitivity to the regularization parameter $\alpha$ that is used. Specifically, we notice that faster decay rates can lead to a severe decline of the initial performance gain achieved when prioritizing learning from teacher-provided actions. Conversely, a slower decay rate appears to work well in these set of environments and allows the agent to obtain expert-level performance within 4 cycles. A possible explanation for this phenomena is that mimicking the teacher's behaviors reduces the need for exploration, which in turn allows the RL agent to (in the background) iteratively improve its critic network. This allows the agent to avoid taking actions that are enforced by an ill-informed (or perhaps overly-optimistic) critic.

To eliminate the dependency on regularization parameters, we examine the performance of utilizing the Q-filter for teacher-provided annotations as expressed in equation 7. Recall, that the Q-filter adaptively determines whether to learn to mimic the teacher's actions or, rather, to optimize the agent's own selected action. While utilizing the Q-filter initially under-performs in comparison to incorporating exponentially decreasing weights, it eventually reaches expert performance in a monotonic fashion without need of hyper-parameter tuning.

**Teacher-Gradient Annotations.** As previously mentioned, we hypothesize that relying on teacher-provided gradients during early periods of policy optimization can circumvent risks associated with learning from a poorly initialized and/or over-estimated value function. Here, we compare policy regularization and teacher-action annotations with using teacher-provided gradient information during policy optimization. We set the gradient information to be $G_a(s, a) = \nabla_a Q^*(s, a)$, i.e., differentiated snapshots of the teacher's internal value function evaluated with respect to the growing-batch agent's current policy. We chose this form due to its easy of use and simplicity. The decay rate for $\alpha$ was set to 1 and 5, respectively. In Figure 6b, we observe that, irrespective of choice in $\alpha$, utilizing gradient information from an expert teacher performs better than solely relying on BC-regularization, but is unable to reach the superior performance gain achieved when leveraging teacher-provided actions. Furthermore, we highlight that without the need of policy regularization leveraging teacher-provided gradients is able to circumvent the initial drop in performance that is observed after pre-training using BC. This insight further supports previously studied hypotheses attributing this anticipated drop in performance to poorly-initialized value functions.

**Sub-optimal Teachers.** In Figure 7, we adapt the previously mentioned approaches by employing a sub-optimal teacher (i.e., an RL agent with mid-tier performance in each task) for both BC initialization and within-cycle training using annotations. Although the growing-batch agent does not reach the same level of performance as when guided by an expert teacher, it surpasses the sub-optimal teacher's average baseline performance in almost all experiments. Importantly, our main conclusions remain consistent despite these changes:

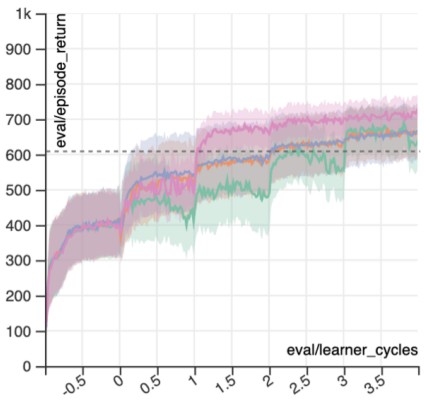

(a) All environments, number of cycles = 4. Baseline using BC-policy regularizer vs. Filtered Teacher-Actions vs. Teacher-Gradient auxiliary loss with decay rate = 1, and decay rate = 5.

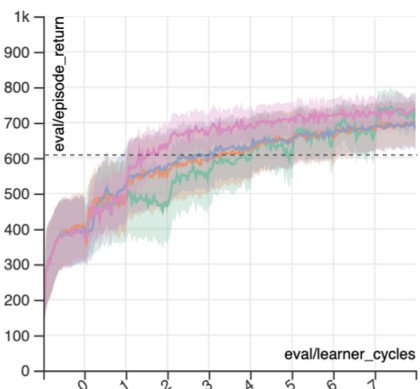

(b) All environments, number of cycles = 8. Baseline using BC-policy regularizer vs. Filtered Teacher-Actions vs. Teacher-Gradient auxiliary loss with decay rate = 1, and decay rate = 5.

Figure 7: Episode return averaged over all tasks (using 5 randoms seeds per task) under teacher-provided annotations. The shaded regions represents the standard deviation across all tasks and random seeds. The dashed line represents the *sub-optimal* teacher's average baseline performance across all tasks.

(a) Employing BC with policy regularization prevents a drastic decline in performance after initialization

(b) Teacher-action annotations, when used with a Q-filter, eventually surpass all other examined methods without requiring hyper-parameter tuning.

(c) Utilizing teacher-gradients (differentiated with respect to actions) results in better performance than relying solely on BC-regularization. Additionally, it helps mitigate the initial drop in performance observed after pre-training, regardless of the chosen decay rate.

## 5    CONCLUSION

In this paper, we present methods to leverage external information from teachers to improve the sample efficiency of growing-batch RL agents, while encouraging safe and monotonic policy improvement. Traditionally, RL agents are trained in an online manner, continuously updating their policy as new data is gathered from environmental interaction. While such approaches have achieved success in low-risk domains such as games, real-world application of RL require extensive evaluation of learned policies before subsequent deployment. As such, training agents in the growing-batch setting operationalizes these desires, while providing a realistic framework for incorporating external information from human experts that serves to improve sample complexity and coverage requirements of conventional RL methodologies.

We demonstrate that while pre-training policies via behavioral cloning can lead to good starting policies, safe optimization using new data is challenging due to overestimation bias present within critic network. Policy regularization can be used to improve this but can also cause the learned policy to stay overly close to the behavioral policy (and data) and limit the overall performance of the agent. We investigate the incorporation of exponentially decaying regularization weights to mitigate the agent's reliance on the behavioral policy as new experience is gathered, improving performance in our of experiments. We further illustrate that external information can also be used during an agents within-cycle training process in the form of transition-specific annotations. In our experiments, we observed that providing expert actions and gradients serves to notably improve the sample efficiency of our agents and encourage monotonic improvement across cycles. Since both types of annotation are practical and feasible, our work provides a suitable framework for further experiments in real-world problems.

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

## A  APPENDIX

### A.1  DESCRIPTION OF ALGORITHM

---

**Algorithm 1:** Teacher-guided Growing-Batch RL

---

**Input:** teacher demonstrations $\mathcal{D}_T$, teacher policy $\pi^*$ and critic-related gradients $G_a(s, a)$,
        batch size $L$, learning rates $\alpha_0$ and $\beta_0$, no. of cycles $N$, actor steps $M$, learner steps $L$

Pre-train policy network $\pi_{\theta_0}$ via behavioral cloning using $\mathcal{D}_T$
Pre-train critic network $Q_{\phi_0}$ via policy evaluation using $\mathcal{D}_T$ and $\pi_{\theta_0}$

**for** $k \in [N]$ **do**
    // **Acting** (Environment Interaction & Data Aggregation):
    **for** $i \in [M]$ **do**
        Sample action $a = \pi_{\theta_{k-1}}(s) + \epsilon \mathcal{N}(0, 1)$
        Execute action $a$, observe $r$ and observation $o'$
        Store transition $\{s, a, r, s'\}$ in replay $\mathcal{D}$
    **end**
    // **Learning** (Within-cycle Policy Optimization):
    **for** $j \in [L]$ **do**
        Sample $B$ transitions from replay buffer $\mathcal{D}$
        Select update procedure $\delta_{\theta_{k-1}}$ for policy network:
        Construct the DPG component $f(s) = \nabla_{\theta_{k-1}} \pi_{\theta_{k-1}} \nabla_a Q_{\phi_k}(s, a)\big|_{a = \pi_{\theta_{k-1}}(s)}$

        **if** *Policy regularization* **then**

$$\delta_{\theta_{k-1}} = \frac{1}{B} \sum_i^B \left( (1 - \alpha_j) f(s_i) + 2\,\alpha_j\, \nabla_{\theta_{k-1}} \pi'_{\theta_{k-1}}(s_i) \left( \pi_{\theta_{k-1}}(s_i) - \pi_{\theta_0}(s_i) \right) \right)$$

        **end**
        **if** *Teacher-action with Q-filter* **then**
            Retrieve teacher action $a^* = \pi^*(s)$
            Construct Q-filter $\delta(s) = \mathbb{1}[Q_{\phi_{k-1}}(s, a^*) \geq Q_{\phi_{k-1}}(s, \pi_{\theta_{k-1}}(s))]$

$$\delta_{\theta_{k-1}} = \frac{1}{B} \sum_i^B \left( [1 - \delta(s_i)] f(s_i) + 2\,\delta(s_i)\, \nabla_{\theta_{k-1}} \pi'_{\theta_{k-1}}(s_i) \left( \pi_{\theta_{k-1}}(s_i) - a^* \right) \right)$$

        **end**
        **if** *Teacher-gradient* **then**
            Retrieve teacher gradient, $G_a(s, \pi_{\theta_{k-1}}(s))$

$$\delta_{\theta_{k-1}} = \frac{1}{B} \sum_i^B \left( (1 - \alpha_j) f(s_i) + \alpha_j\, G_a(s, \pi_{\theta_{k-1}}(s)) \nabla_{\theta_{k-1}} \pi_{\theta_{k-1}}(s) \right)$$

        **end**
        Compute critic update $\delta_{\phi_k} = \frac{1}{B} \sum_i^B \nabla_{\phi_k} \mathcal{L}(\phi_k)$, where $\mathcal{L}(\phi_k)$ is defined in equation 2
        Update policy and critic network $\theta_k \to \theta_{k-1} + \alpha_j \delta_{\theta_{k-1}}, \phi_k \to \phi_{k-1} + \beta_j \delta_{\phi_{k-1}}$
    **end**
**end**
**return** Policy parameters $\theta_N$

---

### A.2  EXPERIMENTAL SETUP

In our experiments, the policy and critic networks have three hidden layers each with 256 units. Under the default D4PG setup, the distribution critic is a neural network layer mapping the output of the critic torso to the parameters of a categorical distribution, i.e., logits $w_i \in (-150, 150)$ defined over a fixed set of 51 atoms $z_i$. The policy networks outputs to a vector whose dimensionality corresponds that of action space of the given task. As the agent acts within the environment, fixed Gaussian noise $\epsilon \mathcal{N}(0, 3)$ is added to the current policy; note that during evaluation no Gaussian noise is added. We use a replay table of size $R = 1 \times 10^7$, where transitions are sampled uniformly. N-step returns are used to construct the distribution TD-error; the trajectory length for these returns are set to $N = 5$. For both actor and critic updates, we initialize the learning rates to $\alpha_0 = \beta_0 = 1 \times 10^4$ and set the

batch size to $M = 256$. The total number of actor steps permitted is set to 2M. Correspondingly, the total number of learner steps is selected such that samples per insert ratio is fixed to 32. Note that samples per insert is a measure of an agent's rate of learning to acting. Specifically, it is measured as the number of samples out of replay per insert into the replay buffer. For a batch size of 256, an SPI of 32 corresponds to a gradient update every 8 actor steps. The number of steps per cycle and learner steps per cycle were determined by dividing the aforementioned totals by the number cycles. We separately experimented with setting the total number of cycles to 4 and 8, respectively.

