# OpenReview forum: "Knowledge Transfer from Teachers to Learners in Growing-Batch Reinforcement Learning"
_ICLR.cc/2023/Workshop/RRL — RRL 2023 Poster_

### Official Review · Reviewer_KHm7 · 2023-03-01
**Lack of baseline comparison**

**Rating:** 2
**Confidence:** 4

**Review:**

Summary:
This paper proposes a scheme to efficiently train a reinforcement learning agent by leveraging knowledge from a teacher policy in a growing-batch setting, where the agent updates its policy only after large batches of data collection. The approach involves the following steps:

1. Pre-train the value function with BC on teacher demonstrations.
2. During policy training, initialize the policy with the pre-trained policy and add a BC regularizer to the policy loss to prevent forgetting the pre-trained knowledge.
3. Assume access to teacher annotations, either teacher-provided actions or gradients and incorporate a regularizer term that uses the teacher annotations to improve sample efficiency.

In the experiment, the authors compared their method with naively fine-tuning the BC initialization and also conducted several ablations to determine the best regularizer and decay rate. The authors showed that their method can achieve sample-efficient fine-tuning.

Strengths:
1. The paper is well-written, and the motivation for studying fine-tuning in the growing-batch setting is clear.
2. The ablations shown in Figures 4 and 5 are interesting and demonstrate the effectiveness of the proposed method compared to the naive BC baseline.

Weaknesses and Open Questions:

1. Lack of baselines: The proposed method is only compared with naively fine-tuning the BC. It would be valuable to compare it with different methods such as using offline RL approaches (e.g., CQL, IQL) or off-policy approaches (e.g., DDPGfD) in each cycle. Additionally, it would be worthwhile to compare it with other prior works that study growing-batch settings (e.g., https://arxiv.org/pdf/2006.03647.pdf).
2. The assumption of the availability of expert gradients is impractical. If the teacher is supposed to be a human annotator in real-world problems, it would be difficult to obtain the gradients.
3. Instead, if the teacher is supposed to be a neural network policy (as in the experiment), other questions arise: if we already have a teacher policy that is near-optimal, why do we need to train another policy? If the teacher policy is sub-optimal, why don't we directly fine-tune the teacher policy?
4. It would be valuable to show the results separately with different levels of teacher optimality and analyze if the learning performance is affected by the teacher's optimality.